# Lipidomic QTL in Diversity Outbred mice identifies a novel function for α/β hydrolase domain 2 (*Abhd2*) as an enzyme that metabolizes phosphatidylcholine and cardiolipin

Tara R. Price[1], Donnie S. Stapleton[1], Kathryn L. Schueler[1], Marie K. Norris[2], Brian W. Parks[3], Brian S. Yandell[4], Gary A. Churchill[5], William L. Holland[2], Mark P. Keller[1], Alan D. Attie[1]*

1 Department of Biochemistry, University of Wisconsin-Madison, Madison, Wisconsin, United States of America, 2 Department of Nutrition and Integrative Physiology, University of Utah, Salt Lake City, Utah, United States of America, 3 Department of Nutritional Sciences, University of Wisconsin-Madison, Madison, Wisconsin, United States of America, 4 Department of Statistics, University of Wisconsin-Madison, Madison, Wisconsin, United States of America, 5 The Jackson Laboratory, Bar Harbor, Maine, United States of America

* adattie@wisc.edu

## Abstract

We and others have previously shown that genetic association can be used to make causal connections between gene loci and small molecules measured by mass spectrometry in the bloodstream and in tissues. We identified a locus on mouse chromosome 7 where several phospholipids in liver showed strong genetic association to distinct gene loci. In this study, we integrated gene expression data with genetic association data to identify a single gene at the chromosome 7 locus as the driver of the phospholipid phenotypes. The gene encodes α/β-hydrolase domain 2 (*Abhd2*), one of 23 members of the ABHD gene family. We validated this observation by measuring lipids in a mouse with a whole-body deletion of *Abhd2*. The *Abhd2^KO^* mice had a significant increase in liver levels of phosphatidylcholine and phosphatidylethanolamine. Unexpectedly, we also found a decrease in two key mitochondrial lipids, cardiolipin and phosphatidylglycerol, in male *Abhd2^KO^* mice. These data suggest that Abhd2 plays a role in the synthesis, turnover, or remodeling of liver phospholipids.

## Author summary

Lipids have broad roles in normal physiology and disruptions to lipid metabolism have been linked to disease development; new roles for the enzymes that metabolize lipids are still being discovered. To identify novel genes associated with a wide array of liver lipids, we conducted a genetic screen of untargeted liver lipidomics in a genetically diverse mouse population. We identified the enzyme, *Abhd2*, as a candidate driver of multiple liver phospholipids. Further, we validated *Abhd2* in a whole-body knockout mouse

**Data Availability Statement:** Raw data for *Abhd2* expression in DO mice and *Abhd2^KO^* mouse

phenotyping data are provided as supporting information. Full DO liver untargeted lipidomic data has been previously published (https://www.nature.com/articles/s42255-020-00278-3).

**Funding:** This work was supported by grants from the NIH (R01DK101573, R01DK102948, and RC2DK125961 (A.D.A.)) and by the University of Wisconsin–Madison, Department of Biochemistry and Office of the Vice Chancellor for Research and Graduate Education with funding from the Wisconsin Alumni Research Foundation (M.P.K.). Research support to T.R.P. was provided through the NIH by the Training Program in Translational Cardiovascular Science (T32-HL007936) at UW Madison. Additional support was provided by the Jackson Laboratory Cube Initiative. The funders had no role in study design, data collection and analysis, decision to publish, or preparation of the manuscript.

**Competing interests:** The authors have declared that no competing interests exist.

model, where loss of *Abhd2* resulted in increased liver phosphatidylcholine and a marked decrease in the mitochondrial lipids, cardiolipin and phosphatidylglycerol. Thus, the genetic screen and *in vivo* validation enabled us to discover a previously unknown role of *Abhd2* in regulating liver phospholipids, including those associated with mitochondrial membranes. Our study highlights the power of lipid genetic screens to nominate and identify novel substrates for enzymes and their larger role in physiology.

## Introduction

Lipids play a variety of roles in physiology, including providing structure, signaling and as fuel sources. Disruptions to lipid metabolism can lead to disease states such as obesity [1, 2], insulin resistance [3, 4], cardiovascular disease [5, 6], and hepatic steatosis [7, 8]. Manipulations to lipid composition in plasma, tissues, and organelles can have a profound impact on disease susceptibility. For example, alterations in the fatty acid compositions of lipids in the endoplasmic reticulum (ER) have been shown to affect obesity-associated ER stress and to improve glucose metabolism in a leptin-deficient mouse model of obesity [9].

Improvements in detection methods and their sensitivity, such as untargeted lipidomics, have allowed for discovery of previously undefined roles of lipids in physiology. Within the past decade, a new class of lipids (fatty acid esters of hydroxy fatty acids, FAHFAs) have been discovered [10]. For example, the identification of FAHFAs as a novel bioactive lipid class has opened a new field of study into their roles in normal physiology and metabolic disease [11–13].

Commensurate with the diversity of lipids is the diversity of enzymes that metabolize lipids. One substantial challenge is discovering the *in vivo* substrates of lipid metabolizing enzymes and the enzymes responsible for synthesis and turnover of newly discovered lipids.

We have used genetics to assist us in establishing a causal link between enzymes and their substrates. When we perform lipidomic surveys in the context of a segregating population, we can identify loci where specific lipid species are genetically associated with loci harboring genes that encode highly plausible candidate enzymes responsible for the metabolism of the lipids. In a prior study, we showed that the substrate and product of an enzyme in glycosphingolipid metabolism mapped to a locus containing that enzyme [14]. This was a proof-of-principle that genetics could be used to de-orphanize lipid metabolism enzymes.

The same study identified several ABHD members as modulators of lipid classes [14]. In validation experiments, ABHD1 and ABHD3 overexpression revealed distinct specificity for lipid classes and acyl chain lengths. The ABHD family of such enzymes (α/β-hydrolase domain) has 23 known members, which are characterized by a α/β-hydrolase fold and a catalytic serine hydrolase domain [15, 16]. ABHD6 is the most characterized lipase in this family, with a wide variety of physiological roles including adipose biology, islet insulin secretion, and cold tolerance [17–20]. ABHD3, another lipase, was shown to selectively modulate phospholipids with C14 acyl chain lengths [21]. The biological roles of many ABHD family members are still being discovered. Here, we incorporate murine liver untargeted, mass spectrometry-based lipidomics and quantitative trait loci (QTL) genetics to identify α/β-hydrolase domain 2 (*Abhd2*) as a novel driver of hepatic phospholipids.

## Results

### Identification of *ABHD2* as novel driver of liver phosphatidylcholine

In a recent genetic screen of circulating and hepatic lipids in Diversity Outbred (DO) mice, we identified a quantitative trait locus (QTL) for multiple phospholipids phosphatidylcholine

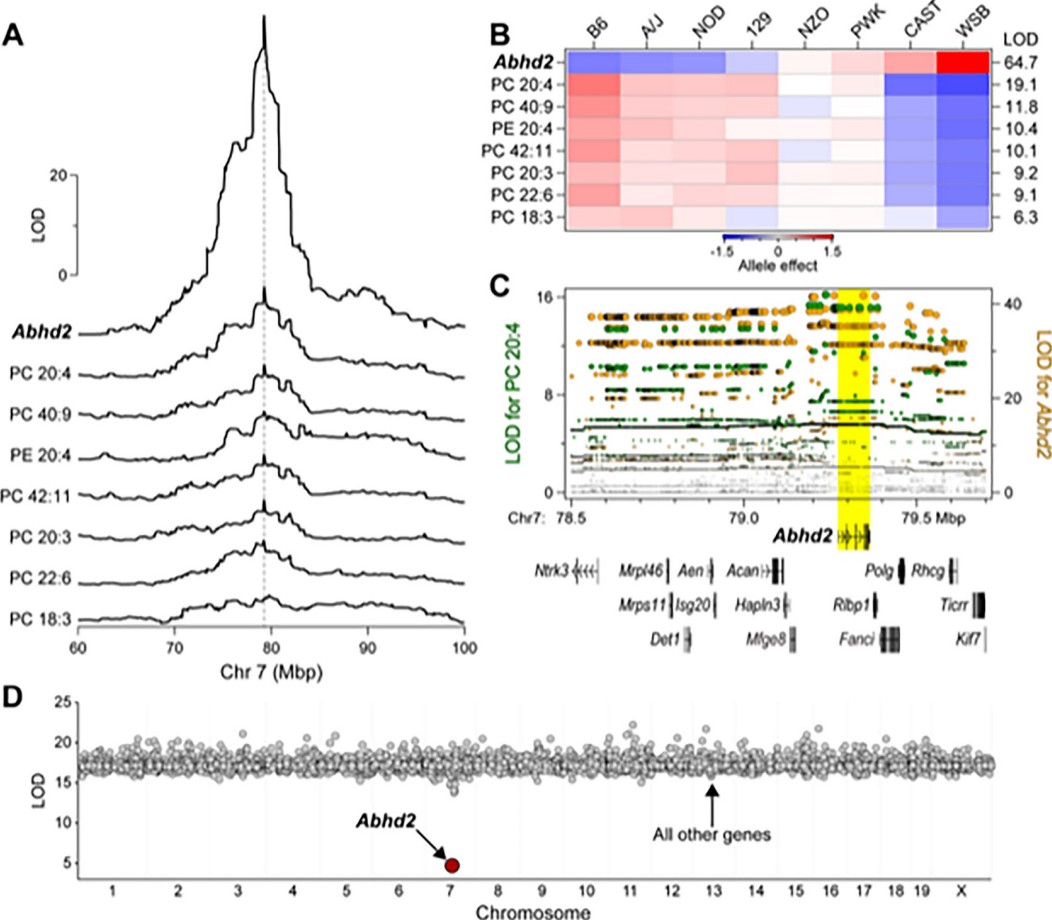

**Fig 1. Identification of *Abhd2* as a driver of phosphatidylcholine (PC) and phosphatidylethanolamine (PE) in liver.** (**A**) LOD profiles for *Abhd2* expression and abundance of several phospholipids in liver identify a common quantitative trait locus (QTL) at ~79 Mbp on chromosome 7. (**B**) Allele effects for the phospholipids and *Abhd2* expression at the chromosome 7 QTL. LOD scores are shown along the right margin and genotypes for the alleles (associated strains) are listed along the top axis. (**C**) Chromosome 7 SNP association profiles for PC 20:4 QTL (left axis) and *Abhd2* eQTL (right axis). All protein-coding genes located between 78.5 and 79.7 Mbp are shown. A block of SNPs with highest association to PC 20:4 and *Abhd2* expression are centered over *Abhd2* gene (yellow highlight). (**D**) Mediation analysis was performed on PC 20:4 QTL by conditioning the lipid QTL on individual genes across the genome. Conditioning on *Abhd2* resulted in the only significant decrease in the LOD for PC 20:4.

(PC) and phosphatidylethanolamine (PE) on chromosome 7 at ~79 Mbp [22]. In parallel, we performed RNA-sequencing to survey the liver transcriptome in the same DO mice that were used for the lipidomic survey, enabling us to identify expression QTL (eQTL) for all genes. We found a strong association of the abundance of the *Abhd2* mRNA with SNPs located near the *Abhd2* gene (a cis-eQTL) with a LOD of 65. This QTL co-mapped with the phospholipid QTL on chromosome 7 (**Fig 1A**).

DO mice segregate alleles from eight founder strains. We can identify the contribution of each allele to a given phenotype and display the allele effect patterns. The allele effect patterns for the phospholipids and the *Abhd2* eQTL were similar, partitioning the founder haplotypes into two subgroups: CAST and WSB versus B6, A/J, NOD and 129 (**Fig 1B**). However, the directionality of the haplotype separation was different for the phospholipids and *Abhd2* expression. Whereas alleles derived from CAST and WSB were associated with high expression

of *Abhd2*, the same alleles were associated with lower abundance of phospholipids (**Fig 1B**). Thus, the phospholipids and *Abhd2* expression show shared but inverted genetic architecture. This inverse pattern is indicative of a "substrate" signature, suggesting that *Abhd2* participates in the phospholipid's degradation.

Next, we identified the SNPs most strongly associated with the phospholipids and the expression of *Abhd2*. The QTL for PC-20:4 peaks at ~79.2 Mbp and includes a block of SNPs with strongest association, which span from ~79.2 to ~79.4 Mbp on chromosome 7 (**Fig 1C**). The gene for *Abhd2* is located ~79.3 Mbp, right under the SNPs with strongest association to PC-20:4. The SNP association profile for the *Abhd2* cis-eQTL was the same as that for PC-20:4, suggesting a common genetic architecture for the lipids and *Abhd2* expression. There are 67 protein-coding and non-coding genes that are located between 78.2 and 80.2 Mbp on Chr 7 (**S1 Table**).

We next used mediation analysis to identify a causal gene driver from among the genes present at the phospholipid QTL. In mediation analysis, the QTL for a lipid is conditioned on the expression of all other genes, including those at the locus to which the lipid maps. If the genetic signal of the lipid QTL decreases upon conditioning of the expression level of a specific gene, that gene becomes a strong candidate as a driver for the lipid QTL. We focused on the QTL for PC-20:4, as this demonstrated the strongest genetic signal (**Fig 1A**). Mediation of the PC-20:4 QTL against the expression of *Abhd2* in liver resulted in a large drop in the LOD score for the PC 20:4 QTL (**Fig 1D**). To extend these observations, we asked if *Abhd2* is a strong driver for all phospholipids mapping to the chromosome 7 QTL. For the seven phospholipids with a QTL to the *Abhd2* gene locus, mediation against *Abhd2* expression resulted in the largest drop in the LOD scores (**S1 Fig**). In summary, the inverse allele effects for the phospholipid versus *Abhd2* expression profiles strongly suggest that *Abhd2* functions as a negative driver of the hepatic phospholipid QTL on chromosome 7.

## Experimental validation of *Abhd2* as a driver of liver phospholipids

To determine if *Abhd2* is a key driver of liver phospholipids, we obtained a whole-body knockout of *Abhd2* from Dr. Polina Lishko at UC Berkeley [23, 24]. Wildtype (WT) and *Abhd2* knockout (*Abhd2^KO^*) mice were maintained on the same Western diet (WD), high in fat and sucrose, that was provided to DO mice used for the lipidomic genetic screen [14].

To experimentally validate the genetic prediction that *Abhd2* is a driver of liver phospholipids (PC and PE), we performed mass spectrometry (MS)-based lipidomics on liver tissue from male and female WT and *Abhd2^KO^* mice. A total of 583 unique lipid species were quantified (**S2 and S3 Tables**), including 67 and 50 PC and PE lipids, respectively. **Fig 2** highlights the liver lipids that were the most differentially abundant between WT and *Abhd2^KO^* mice. Female *Abhd2^KO^* mice had 21 liver lipids decreased and 9 lipids increased (**Fig 2A**), whereas male *Abhd2^KO^* mice showed 44 and 16 liver lipids decreased and increased, respectively (**Fig 2B**). Consistent with the prediction from the genetic screen, the PC and PE species that mapped to the chromosome 7 QTL were significantly increased in liver from both male and female *Abhd2^KO^* mice (**Fig 2C**).

In addition to PC and PE, other lipids were significantly altered in the liver of *Abhd2^KO^* mice. For example, several species of cardiolipin (CL) (**Fig 2D**) and phosphatidylglycerol (PG) (**S2A Fig**) were significantly reduced in liver from male, but not female, *Abhd2^KO^* mice. CL and PG are synthesized in mitochondria[25] and play important roles in mitochondrial function [26]. To determine if the decrease in CL and PG levels in *Abhd2^KO^* males reflect a change in mitochondrial number, we performed quantitative PCR for several mitochondrial-encoded genes. In both male and female *Abhd2^KO^* mice, the expression of eight mitochondrial-encoded

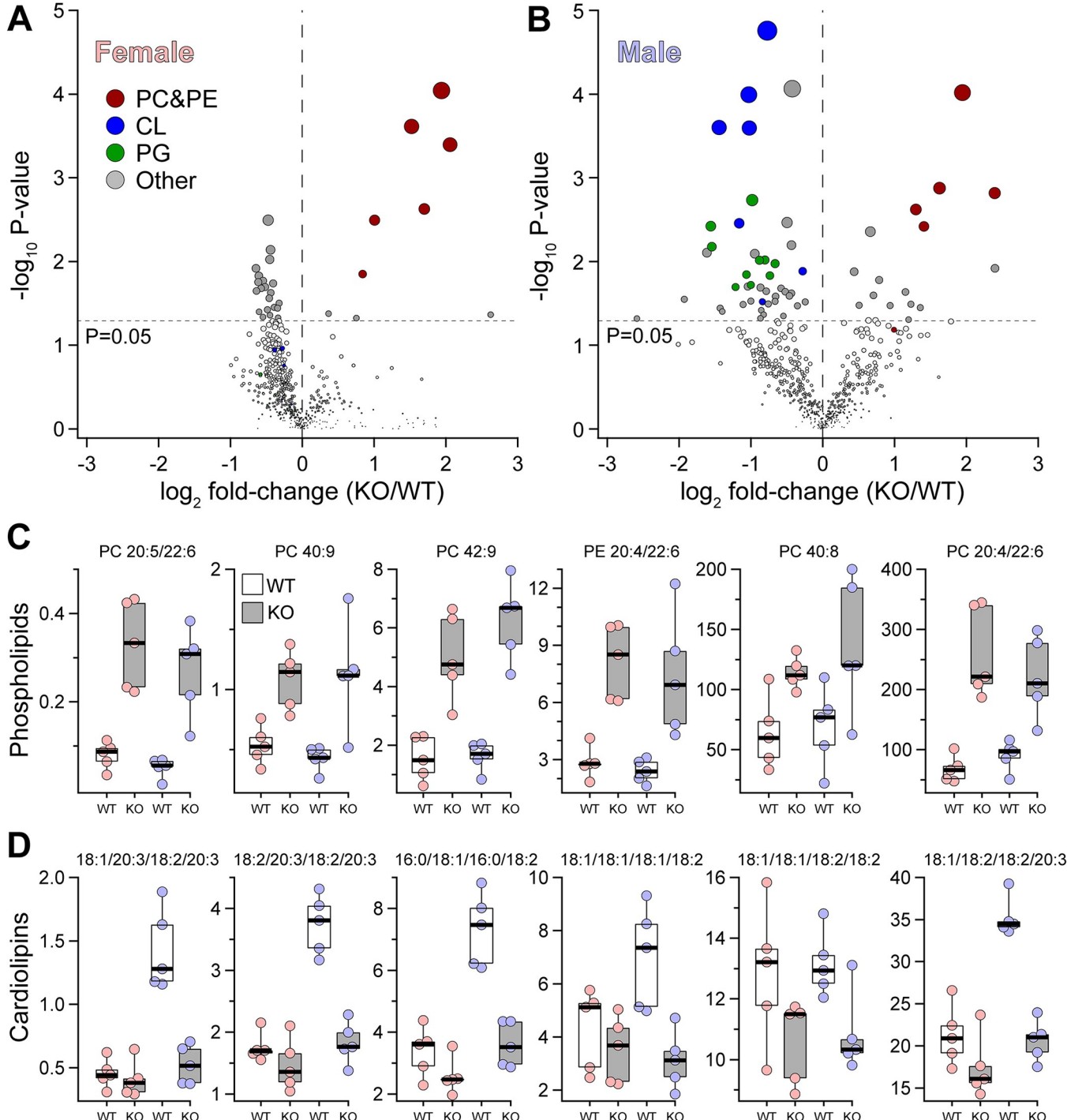

**Fig 2. *Abhd2^{KO}* have reduced levels of hepatic lipids predicted from lipidomic genetic screen.** MS-based lipomics was used to survey the level of ~580 lipids in liver of WT and *Abhd2^{KO}* mice. A total of 29 and 60 lipids were differentially abundant in female (**A**) and male (**B**) *Abhd2^{KO}* mice. Specific lipid classes (PC and PE, CL and PG) are indicated by color. In both female and male mice, several phospholipid species were increased in *Abhd2^{KO}* mice (**C**). These same lipids demonstrated a QTL to the *Abhd2* gene locus on chromosome 7. Male *Abhd2^{KO}* mice have a significant decrease in seven cardiolipin (CL) species (**D**). PC and CL data are presented as pmol lipid per mg liver tissue.

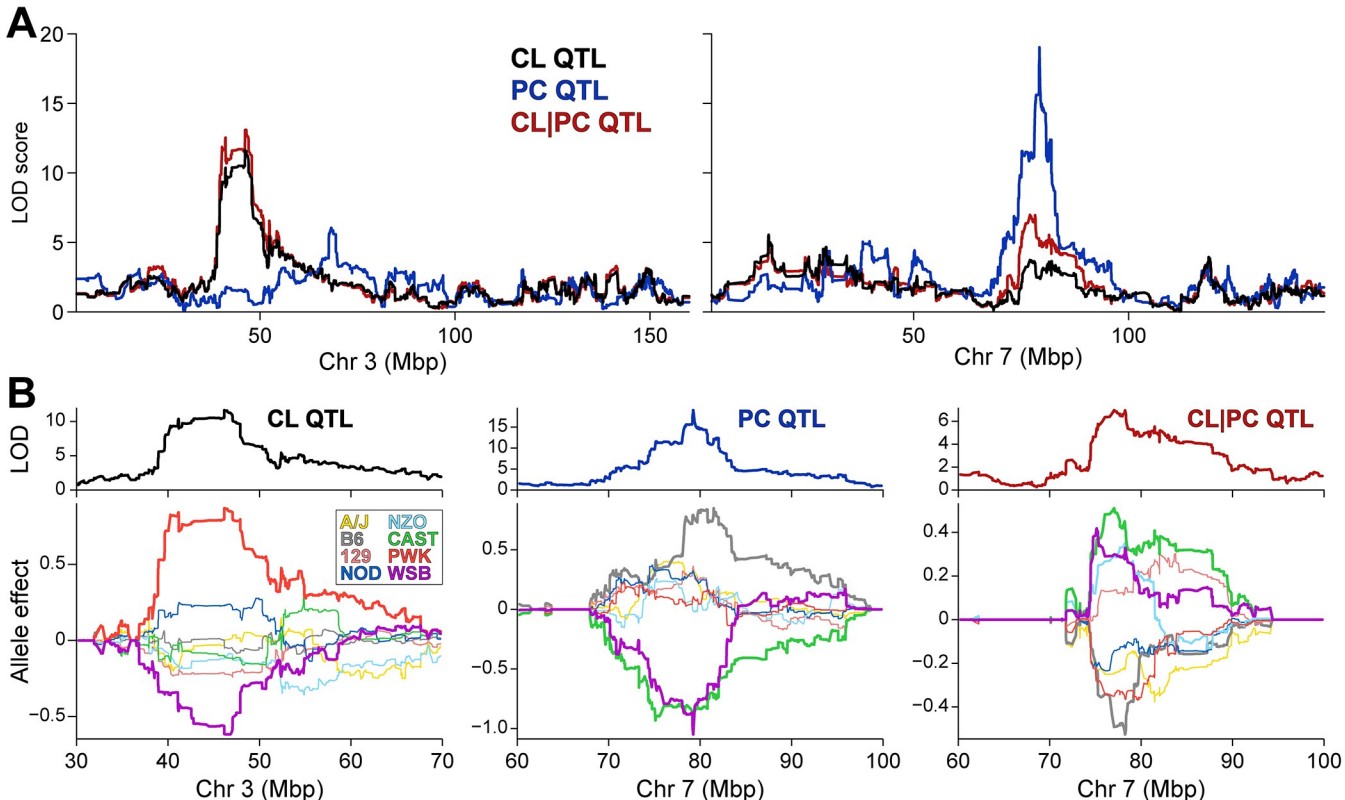

**Fig 3. A cardiolipin hotspot on chromosome 3 is associated with a phosphatidylcholine hotspot on chromosome 7.** (**A**) Genome-wide LOD profile of liver PC (20:4/22:6, blue) identified a QTL on chromosome 7 at ~79 Mbp. Genome-wide LOD profiles for CL (16:0/18:1/16:0/18:1) without (tan) and with (red) conditioning on PC (20:4/22:6) identified QTL on chromosomes 3 at ~46 Mbp and 7 at ~79 Mbp, respectively. (**B**) Allele effects of CL (left), PC (middle), and CL conditioned on PC as an additive covariate (right, denoted CL|PC). CL and PC show distinct allele effect pattern; however, CL conditioned on PC shows a similar, but inverse pattern to PC.

genes was not significantly different in WT *vs. Abhd2^{KO}* mice (**S2B Fig**). These results suggest that lower levels of hepatic CL and PG in male *Abhd2^{KO}* mice are not the consequence of reduced mitochondrial number. It is therefore more likely that ABHD2 plays a key role in the metabolism of these two mitochondrial lipids.

To provide additional support for ABHD2 in regulating hepatic PG and CL levels, we asked if there was genetic association for CL and PG lipid species in liver among DO mice. We identified several QTL for both lipids, including a hotspot on chromosome 3 at ~46 Mbp where several CL species co-mapped (**S4 Table**). CL-16:0/18:1/16:0/18:1 yielded the strongest genetic signal on chromosome 3, with a LOD of ~12, along with possible secondary QTL on chromosomes 7 and 13 (**Fig 3A**). Interestingly, the gene *Abhd18*, which is relatively uncharacterized but has been localized to mitochondria [27], is physically located at the CL QTL on chromosome 3, raising the possibility that *Abhd18* and *Abhd2* work in concert to regulate CL levels. While no CL species mapped to the *Abhd2* gene locus on chromosome 7, conditioning CL on PC-20:4/22:6 as an additive covariate resulted in CL acquiring a QTL to the *Abhd2* locus (**Fig 3A**). This QTL on chromosome 7 of CL adjusted by PC demonstrates an allele pattern that is like that of the cis-eQTL for *Abhd2*, and the inverse of the PC QTL (**Fig 3B**), consistent with CL being a downstream product of ABHD2-dependent metabolism of PC. Similar results were observed for two PG lipids; when conditioned on PC-20:4/22:6, QTL were acquired to the *Abhd2* gene locus (**S5 Table**).

Changes in fatty acyl composition (number of carbons and degree of saturation) have been associated with differential response to metabolic stressors [28, 29]. Therefore, we evaluated the composition of the acyl chains in PC, PG, and CL lipids in WT and *Abhd2*$^{KO}$ mice (**S2C–S2E Fig**). Both PC and PG lipid classes were equally represented by acyl chain lengths of C16 and C18; in CLs, however, C18 comprised more than 95% of the acyl chains (**S2C–S2E Fig**). PCs were primarily composed of saturated fatty acids, PGs had similar monounsaturated and saturated fatty acyl chains (~43% and 50%, respectively), while 80% of CL fatty acyl chains contained two double bonds (**S2C–S2E Fig**). Acyl chain length and degree of saturation for PC, PG, and CL species were not different in *Abhd2*$^{KO}$ mice. Taken together, these results suggest that ABHD2 is not involved in specific alteration of the acyl chain composition of phospholipids.

## Physiological characterization of *Abhd2*$^{KO}$ mice

While the increase in hepatic phospholipids we observed in the *Abhd2*$^{KO}$ mouse confirms the predictions from the genetic screen that *Abhd2* is a negative driver of these lipids, it does not inform us about the physiological role of ABHD2. To gain a better understanding of this, we performed a series of physiological measurements in WT and *Abhd2*$^{KO}$ mice.

WT and *Abhd2*$^{KO}$ mice demonstrated similar growth curves (**S3A and S3E Fig**) and comparable fasting glucose, insulin, and triglyceride (TG) profiles (**S3B–S3D** and **S3F–S3H Fig**). At ~24 weeks of age, body weight did not differ among female mice (WT 32 ± 1g *vs.* Abhd2$^{KO}$ 36 ± 12g, **S4A Fig**)**;** however, *Abhd2*$^{KO}$ females showed greater fat mass (**S4B Fig**), greater fat mass percentage (**S4C Fig**) and decreased lean mass percentage (**S4E Fig**). Male mice did not differ in body weights or body composition (**S4F–S4J Fig**).

To evaluate a role for *Abhd2* deletion on broad metabolic pathways, we performed an oral glucose tolerance test (oGTT) to assess whole-body insulin signaling and glucose homeostasis, a β$_3$-adrenergic receptor agonist tolerance test (β$_3$TT) to examine differences in adipose lipolysis and glucose metabolism, and a fast/re-feed (FRF) paradigm to probe liver lipolysis/lipogenesis pathways.

During the oGTT, no differences in plasma glucose, insulin, or C-peptide levels were observed for male or female WT *vs Abhd2*$^{KO}$ mice (**S5 Fig**). Administration of CL-316,243 (a β$_3$-adrenergic receptor agonist) resulted in a marginal increase in plasma glucose in male *Abhd2*$^{KO}$ mice during the β$_3$TT (**S6 Fig**). However, area under the curve (AUC) for glucose, insulin, free fatty acids, and glycerol were all unchanged in *Abhd2*$^{KO}$ mice (**S6 Fig**). Similarly, circulating fatty acids were not different for WT vs. *Abhd2*$^{KO}$ mice during the fast/re-feed paradigm (**S7 Fig**). Another member of the ABHD family of enzymes, ABHD6, has been shown to have a direct effect on islet insulin secretion by hydrolyzing monacylglycerols, inhibiting MUNC13-1 action and thereby regulating insulin granule release [20]. To directly evaluate the effect of ABHD2 on pancreatic β-cell function, we determined insulin secretion from cultured islets isolated from WT and *Abhd2*$^{KO}$ mice. Insulin secretion in response to varying glucose concentrations or monoacylglycerol (2-arachidonoylglycerol or 1-palmitoylglycerol) was the same for WT and *Abhd2*$^{KO}$ mice (**S8 Fig**).

Given that hepatic phospholipids have been shown to play a major role in lipoprotein metabolism and cholesterol homeostasis [30–33], we measured circulating total cholesterol and triglycerides (TG) in WT and *Abhd2*$^{KO}$ mice. Total cholesterol and TG were not different in *Abhd2*$^{KO}$ mice (**S9A and S9B Fig**). To assess whole-body cholesterol metabolism, we measured biliary and hepatic cholesterol content. These remained unchanged in *Abhd2*$^{KO}$ mice (**S9C Fig**). Hepatic cholesterol levels showed a marginal increase in male, but not female *Abhd2*$^{KO}$ mice (**S9D Fig**).

To assess lipoprotein classes (e.g., LDL, HDL), we performed fast protein liquid chromatography (FPLC) on plasma from WT and *Abhd2*$^{KO}$ mice. Cholesterol in the individual FPLC fractions did not differ between genotypes of females (**S9E Fig**) or males (**S9F Fig**). No differences were detected for total cholesterol across the lipoprotein fractions for WT vs. *Abhd2*$^{KO}$ mice (**S9G Fig**).

Given the marginal increase in hepatic cholesterol levels in male *Abhd2*$^{KO}$ mice (**S9D Fig**), we evaluated hepatic LDL receptor (LDLR) protein levels by western blot analysis. LDLR protein was not different between female (**S9H Fig**) or male (**S9I Fig**) WT and *Abhd2*$^{KO}$ mice (**S9J Fig**). Taken together, while our data supports *Abhd2* as a driver of several hepatic phospholipid and cardiolipin (in male) species, we were unable to link these changes to differences in serum lipoproteins, suggesting that the role of ABHD2 in phospholipid metabolism is confined to intracellular lipids.

## Discussion

Genetic diversity plays a pivotal role in lipid metabolism and homeostasis. By leveraging genetic diversity of murine populations, it is possible to define novel drivers of physiological traits, including lipid classes.

Through untargeted MS-based lipidomics in the context of a genetic screen, our study is the first to nominate and validate *Abhd2* as a genetic driver of hepatic phosphatidylcholine and phosphatidylethanolamine. Phospholipid species (PC and PE) that mapped to chromosome 7 were increased in livers of knockout mice (both sexes), following the substrate signature prediction of our genetic screen. By integrating lipidomics and transcriptomics, we show how a mouse genetic screen can be used to identify novel drivers of hepatic lipids.

*Abhd2* has been previously characterized as a monoacylglycerol lipase with potent effects on male fertility [24] and ovulation in female mice [23]. In sperm, ABHD2 is activated by progesterone and cleaves monoacylglycerols (1-arachadonoylglycerol and 2-arachadonoylglycerol) to remove the inhibition of the CatSper calcium channel, thereby allowing for sperm activation. In a gene-trap mouse model of age-related emphysema, loss of *Abhd2* resulted in decreased PC levels in bronchoalveolar lavage [34]. These *Abhd2*-deficient mice had increased lung macrophage infiltration and inflammatory markers and spontaneously developed emphysema with aging. It is interesting that their study showed a decrease in PC lipids with loss of *Abhd2*, whereas PCs increased in livers of our whole-body *Abhd2*$^{KO}$ mice, perhaps highlighting tissue-specific roles of ABHD2. Nevertheless, *Abhd2* appears to have a causative role in PC species homeostasis. Our study is the first to demonstrate an *in vivo* role for *Abhd2* in phospholipid regulation in non-reproductive tissues.

An unexpected finding was a decrease in cardiolipins and phosphatidylglycerols in male *Abhd2*$^{KO}$ mice. Cardiolipins comprise ~20% of the inner mitochondrial membrane, whereas phosphatidylglycerols reside in the outer mitochondrial membrane [26]. To explore a genetic association between PC and CL or PG, we performed QTL analyses in which the PC lipid showing strongest association to the *Abhd2* gene locus (PC-20.4/22.6) was used as an additive covariate when mapping CL or PG. This QTL analysis yielded an intriguing result: CL and PG acquired QTL at the *Abhd2* locus with an inverted allele signature to that for the PC. This inverted allele signature is also indicative of a substrate signature, where an increase in PC is associated with a decrease in PG and CL. Thus, ABHD2, through its effect on PC, may indirectly play a role in the synthesis of CL species.

One hypothesis for ABHD2's effect on CL biosynthesis is through the role of an acyltransferase. ABHD2 contains two enzymatic motifs: the canonical serine hydrolase motif and the highly conserved HxxxxD acyltransferase motif between H120 and D125. Synthesis of CL

involves a transfer of a fatty acyl chain from PC or PE phospholipids to monolysocardiolipin (MLCL) to form mature CL species. Four MLCL species were detected in our liver samples (S3 Table). In males, there was a 2.5-fold reduction in one MLCL species (MLCL-56:6) in *Abhd2*<sup>KO</sup> mice. If ABHD2 affected mature CL synthesis through a direct fatty acyl chain transfer to MLCL, an increase in MLCL species would be expected. Therefore, the reduction in MLCL indicates ABHD2's role is likely upstream of mature CL synthesis. Since PG is also required for CL synthesis, it's also possible that the reduction in CL concentrations is secondary to alterations in PG concentrations [25]. In our initial QTL analyses of all liver lipids, we identified a CL hotspot on chromosome 3 at ~46 Mbp, which includes the ABHD enzyme, *Abhd18*. Recently, ABHD18 was shown to reside in the mitochondria [27]; however, its mechanism has not been well characterized. In the STRING protein-protein association network database (string-db.org), ABHD2 and ABHD18 are predicted to have an interaction, although this has not been experimentally validated [35]. It is possible that ABHD2 mediates utilization of PC or its acyl chains in the synthesis of CL and PG, or that it interacts with another mitochondrial enzyme, such as ABHD18, to effect these changes.

It is important to note that changes to mitochondrial lipids were only observed in male *Abhd2*<sup>KO</sup> mice, whereas the increase in PC and PE phospholipids occurred in both sexes. Progesterone-induced activation of ABHD2 is required for its lipid cleavage function and regulating ovulation in females [23, 24]; however, the effect of male sex hormones on ABHD2 has not been demonstrated. In a study of cerebral cortex development, a perinatal testosterone spike in male mice drove mitochondrial lipid composition and maturation [36]. It is possible that ABHD2 is required for testosterone-dependent regulation of mitochondrial lipid synthesis or maturation.

Reduced abundance of PG and CL lipids may indicate a reduction in total mitochondrial number or a defect in the inner mitochondrial membrane leading to altered metabolic function. As a surrogate for mitochondrial number, we measured expression of key mitochondrial genes by qPCR but did not see a sex-specific or genotype effect. Thus, the decrease in CL and PG does not appear to be due to a reduction in mitochondrial number but does not rule out altered mitochondrial function in liver from *Abhd2*<sup>KO</sup> mice.

The monoacylglycerol lipase, *Abhd6*, has also been shown to modulate mitochondrial lipid metabolism [37, 38]. However, the changes in lipid class concentrations were in the opposite direction of the *Abhd2* lipids. Loss of *Abhd6* results in an increase in liver PG, which was attributed to defective degradation of lysophosphatidylglycerol (LPG) [37]. Another group later showed increased plasma concentrations of bis(monoacylglycerol)phosphate (BMP) in mice lacking *Abhd6* and in humans with a loss-of-function mutation in *ABHD6* [38]. Both BMP and CL synthesis require PG as a precursor [39]; therefore, it is possible that the reduction of PG and CL content in the *Abhd2*<sup>KO</sup> livers may reflect alterations in one or both of these pathways. Recently, the *Abhd2* locus was linked to age-related macular degeneration (AMD) through a human GWAS of mitochondrial variants [40]. Alterations to mitochondrial function and lipid oxidation have been implicated in AMD disease progression [41, 42]. We did not assess phenotypes related to vision; however, it would be intriguing to determine a role for *Abhd2* in mitochondrial function and AMD disease risk. We observed a reduction in CL in livers of *Abhd2*<sup>KO</sup> mice; thus, we are tempted to speculate that *Abhd2* deficiency may contribute to macular degeneration through its effects on CL.

Loss of *Abhd2* has been previously shown to regulate vascular smooth muscle migration and induce blood vessel intima hyperplasia after a cuff experiment in a mouse model [43]. The same group showed an increase in macrophage *ABHD2* expression abundance in vulnerable plaques in humans [44] but no mechanism of action was determined.

In human genome-wide association studies [45], there is a significant region on chromosome 15 associated with coronary artery disease (CAD). This locus sits between two genes: *ABHD2* and *MFGE8*. Soubeyrand *et al.* showed that deletion of the intergenic locus results in a marked increase in *MFGE8* expression but did not affect the expression of *ABHD2* [46]. Knockdown of *MFGE8* in coronary smooth muscle cell and monocytes inhibited proliferation, indicating *MFGE8* as the causal gene for CAD-associated at this locus [46]. Splice variants of *MFGE8* have been associated with reduced risk of atherosclerosis in FinnGen, a large Finnish biobank study [47]. However, an *in vivo* role for MFGE8 has not been established. In our genetic screen, hepatic expression of *Mfge8* did not significantly correlate with hepatic lipids or plasma lipoproteins. We did not observe a difference in plasma lipoproteins with *Abhd2* deletion. We did not assess any indicators of vascular smooth muscle physiology or blood pressure. Thus, *ABHD2* is likely not the causative gene at the CAD-associated region on chromosome 15 in human GWAS.

All studies were completed in whole-body knockout mice, allowing us to definitively identify *Abhd2* as the driver of the observed phenotypes. For this reason, one limitation of our study is that we cannot speak to tissue-specific roles of *Abhd2* on metabolic stress and disease risk. It will be important for future studies to carefully consider tissue-specific knockout models to further refine our understanding of *Abhd2's* role in physiology. Similarly, we did not assess effects of *Abhd2* at the subcellular level. Disruptions in phospholipid homeostasis could lead to subcellular dysfunction, such as those seen in lysosomal storage disorders (LSD). We did not observe sphingomyelin or cholesterol accumulation in liver nor evidence of muscle wasting, which are often hallmarks of LSD. However, we did not perform microscopy of lysosomes to directly assess alterations to morphological features and our untargeted lipidomics did not annotate bis(monoacylglycerol)phosphates, the key lipids of lysosomes. Additionally, we did not expand on ABHD2's known mechanisms, namely that it operates as a lipase to cleave monoacylglycerols [24] and also contains an acyltransferase motif [15]. Instead, we identified new substrates for ABHD2 and its potential roles in phospholipid homeostasis.

With biochemical approaches alone, it is challenging to discover novel candidate substrates for known enzymes. Through integration of gene expression data with untargeted, mass-spectrometry lipidomics, we identified a hepatic phospholipid hotspot on chromosome 7 and nominated *Abhd2* as a novel driver of PC, PE, and cardiolipin. Using a whole-body knockout mouse model, we validated *Abhd2* as the causative gene for several PC and PE lipids, and CLs, precisely as predicted by the QTL analysis. Our study demonstrates the power of metabolite QTL analysis to discover novel candidate substrates for enzymes.

## Methods

### Ethics statement

All animal work was approved by the Institutional Animal Care and Use Committee at University of Wisconsin-Madison under protocol #A005821.

### Mouse genetic screen to nominate novel drivers of hepatic lipid metabolism

Details of the mouse genetic screen has been previously described [22]. Briefly, 500 Diversity Outbred (DO) mice were obtained from Jackson Laboratories (Bar Harbor, ME) and maintained on a high-fat, high-sucrose diet (TD.08811, Envigo, Madison, WI) for 16 weeks. For this study, livers from 384 mice (191 female, 193 male) were collected for transcriptomics and untargeted mass spectrometry-based lipidomics. Mapping of gene expression and phenotypes

were performed to identify quantitative trait loci (QTL) and nominate candidate drivers for individual lipid species using the GRCm38 genome build and Ensembl 75 for gene annotation. Genome scans were completed with R/qtl2 software [48], using sex and wave as additive covariates. To investigate genetic associations between mitochondrial lipid classes and phosphatidylcholines mapping to chromosome 7, genome QTL scans were performed with sex, wave, and PC-20:4/22:6 as additive covariates. A logarithm of odds (LOD) greater than 6.0 was used as the threshold for identifying suggestive QTL and LOD greater than 7.5 identified significant QTL. As previously described, LOD thresholds were defined through permutation testing to establish a genome-wide family-wide error rate (FWER) for genome-wide QTL [14, 49]. Mediation analysis to establish causality was performed by regression of the target phenotype on the locus genotype to establish the direct genetic effect [50]. Next, we included the candidate gene expression as a covariate in the regression. If the phenotype-genotype association was no longer significant in the conditional regression model, we considered the gene to be a mediator of the genetic effect on the target locus.

### Abhd2 mouse housing and maintenance

Whole-body *Abhd2* heterozygous mice, generated on a C57BL/6N background [23], were a kind gift of Dr. Polina Lishko at University of California–Berkley. All animal work was approved by the Institutional Animal Care and Use Committee at University of Wisconsin-Madison under protocol #A005821. Heterozygous mice were backcrossed with C57BL/6J mice and bred to produce knockout mice and wild-type littermate controls. All mice were housed at the University of Wisconsin–Madison animal facilities with standard 12-hour light/dark cycles. Animals were weaned and provided a high-fat, high-sucrose diet (TD.08811, Envigo, Madison, WI) and water *ad libitum*. At 23–25 weeks of age, mice were euthanized by carbon dioxide asphyxiation and exsanguinated by cardiac puncture. Whole blood was collected with EDTA, centrifuged at 10,000xg for 10 minutes at 4°C and plasma separated. Tissues were collected, snap frozen in liquid nitrogen, and stored at -80°C until assay.

### *in vivo* physiologic measurements

At 6, 10 and 14 weeks of age, mice were fasted four hours and blood collected by retro-orbital bleed for measurement of plasma glucose (#23-666-286, FisherScientific), insulin (#SRI-13K, MilliporeSigma) and triglycerides (#TR22421, ThermoFisher). At age 16 weeks, mice were subjected to a 24-hour fast and 6-hour refeed to assess hepatic lipid storage during energy deficits. Body weights and whole blood were collected at 0, 24, and 30 hours and plasma measured for non-esterified fatty acids (NEFA) using the Wako Linearity Set (#999–34691, #995–34791, # 991–34891, # 993–35191, FisherScientific). *in vivo* insulin action was assessed at 18 weeks of age by an oral glucose tolerance test as previously described [22]. Mice were fasted for four hours and given a 2 g/kg BW glucose dose by oral gavage. Blood was collected by retroorbital eye bleed and assayed for glucose, insulin, and c-peptide concentrations. $\beta_3$-adrenergic receptor agonist tolerance tests ($\beta_3$TT) were performed at 20 weeks of age on four-hour fasted mice. Mice were dosed with 1 mg/kg BW of CL-316,243 by i.p. injection. Blood, collected by retroorbital eye bleed, was assayed for glucose, non-esterified fatty acids, glycerol, and insulin content.

### Liver lipidomics

Frozen tissues were sectioned to 10mg on dry ice and added to phosphate buffered saline (PBS) and methanol containing internal stable isotope metabolomics standards (S6 Table). Tissues were mechanically homogenized (Qiagen TissueLyser) for 5 minutes at maximum

frequency (30.0 Hz/s). 20μL of homogenate was removed for protein quantification (Pierce BCA Protein Assay Kit). Samples were mixed with methyl tertiary-butyl ether (MTBE), vortexed, centrifuged, and supernatant was transferred into new tube. Original samples were re-extracted with MTBE: Methanol: dd-H2O (10:3:2.5), vortexed, centrifuged, and supernatant was transferred into tubes containing the first extraction's supernatant. Samples were evaporated in a speed-vac and then resuspended with isopropyl alcohol: acetonitrile: dd-H$_2$O (8:2:2). Samples were then vortexed and centrifuged before transferring supernatant to glass vials (Agilent Technologies). Samples were analyzed by liquid chromatography- tandem mass spectrometry (LC-MS) with a 6545 UPLC-QToF mass spectrometer for non-targeted lipidomics. Results from LC-MS experiments were collected using Agilent Mass Hunter Workstation and analyzed using the software package Agilent Mass Hunter Quant B.07.00. Lipid species were quantified based on exact mass and fragmentation patterns and verified by lipid standards. Mass spectrometry was performed at the Metabolomics Core Facility at the University of Utah. Mass spectrometry equipment was obtained through NCRR Shared Instrumentation Grant 1S10OD016232-01, 1S10OD018210-01A1 and 1S10OD021505-01.

## Liver, bile, and plasma cholesterol

Total cholesterol in undiluted plasma and bile was assessed with Infinity Cholesterol reagent (TR13421, Thermo Scientific, Waltham, MA) and concentrations determined by a standard curve. Liver cholesterol was extracted by homogenizing 50 mg of tissue in a TissueLyser with 1 mL chloroform:isopropanol:IGEPAL CA-630 (7:11:0.1). The organic phase was collected and dried at 50°C. Dried lipids were resuspended in 200μL cholesterol assay buffer (MAK043, Millipore Sigma, St. Louis, MO) and total cholesterol determined following manufacturer's protocol.

To analyze lipoprotein size distributions, plasma was analyzed using a Superose 6 10-300GL column and size-exclusion fast protein liquid chromatography (FPLC). Fractions were assayed for total cholesterol and triglycerides as previously described [51].

## RT-PCR for mitochondrial genes

For mitochondrial gene analyses, DNA was isolated from liver samples (n = 5/sex/genotype) with an overnight incubation in proteinase K. Isolated DNA was dried and resuspended in ultrapure water for qPCR analysis. Mitochondrial gene expression (primers in S7 Table) were normalized to the nuclear cystic fibrosis transmembrane conductance receptor (*Cftr*) and fold-change calculated using the $2^{-\Delta\Delta Ct}$ method.

## Western blot analysis

Tissues were lysed in RIPA buffer and total protein determined by Pierce BCA assay (#23225, ThermoFisher Scientific) to ensure equal loading. Samples (15–30 ug) were heat inactivated with 4X Laemmli dye containing 4% 2-mercaptoethanol at 70°C for 10 minutes and run on 7.5% tris-glycine gels following standard protocols. PVDF membranes were stained for total protein with 0.1% ponceau S in 5% acetic acid, and then probed for the protein of interest. For blotting of FPLC-separated plasma lipoprotein fractions, 25 μL of each fraction was incubated with 4X Laemmli dye containing 4% 2-mercaptoethanol at 70°C for 10 minutes and probed for protein as described above. Primary and secondary antibodies are listed in S8 Table.

## Statistical analyses

Statistical analysis of *in vivo* mouse data and tissue assays were performed by ANOVA followed by Tukey's post-hoc analysis. Lipidomics data were analyzed using MetaboAnalystR

[52]: liver lipid concentrations (pmol lipid/mg liver) were $\log_{10}$-transformed, normalized by Pareto scaling, and then fold change calculated. Unless noted, data are presented as mean ± standard error. Differences were considered significant at p<0.05.

## Statement of data availability

Raw data for *Abhd2* expression in 500 DO and *Abhd2*^KO^ mouse phenotyping data are provided as supporting information. DO liver lipidomic data has been previously published [14].

## Supporting information

**S1 Fig. Mediation analysis of phospholipid QTL identifies *Abhd2* as candidate causal gene.** Mediation analysis of the QTL for seven liver lipids that map to the chromosome 7 locus resulted in substantial LOD drop when conditioned on hepatic *Abhd2* expression. Conditioning on all other genes did not result in any appreciable LOD drop for these lipids.
(TIF)

**S2 Fig. *Abhd2* deletion decreased hepatic phosphatidylglycerol concentrations but did not alter mitochondrial gene expression or mitochondrial lipid acyl chain compositions.** (**A**) Despite significant reductions in several cardiolipin species in male mice, total hepatic cardiolipin levels in male and female mice did not differ by genotype. However, total phosphatidylglycerol concentrations were decreased Abhd2^KO^ mice compared to WT males (p<0.01). (**B**) Mitochondrial gene expression, measured as a proxy for mitochondrial number, was not different by sex or genotype. Neither genotype nor sex affected fatty acyl composition of PC, PG or CL in the livers of HF/HS-fed mice. (**C**) The hepatic phosphatidylcholine landscape was diverse and primarily comprised of acyl chains of C16 or C18 in length and were saturated or monounsaturated. (**D**) Phosphatidylglycerols were equally represented by fatty acyl lengths of C16 and C18 and contained 0 or 1 double bond. (**E**) Cardiolipins were highly represented by linoleate, with C18 being 95% of acyl lengths and 98% of CLs containing 1 or more double bonds.
(TIFF)

**S3 Fig. Whole-body deletion of *Abhd2* did not alter growth rates nor fasting blood profiles in C57BL/6J mice.** Abhd2^KO^ female (**A**) and male (**E**) mice showed similar growth curves to WT mice. Fasting glucose (**B, F**), insulin (**C, G**), and triglycerides (**D, H**) did not differ by genotype.
(TIFF)

**S4 Fig. Loss of *Abhd2* altered body compositions of female mice by increasing fat mass as measured by DEXA.** Body compositions of mice were measured at ~24 weeks of age by DEXA. (**A**) Body mass of female mice were not significantly different. Fat mass, both as total weight (**B**) and %body weight (**C**) increased in *Abhd2*^KO^ female mice. Lean mass weight (**D**) did not change with genotype in females, but lean mass as %body weight (**E**) was reduced in *Abhd2*^KO^ female mice. Male mice were not different in total body weight, fat, nor lean mass (**F-J**). *p<0.05.
(TIFF)

**S5 Fig. Assessment of insulin action by oral glucose tolerance test (oGTT) elicited similar responses between genotypes of the same sex.** (**A**) Female *Abhd2*^KO^ mice showed a trend for increased plasma glucose at 15 and 30-minute timepoints during the oGTT. Male *Abhd2*^KO^ mice were not different. (**B**) Area under the curve (AUC) for plasma glucose during the oGTT did not differ by genotype. (**C**) Plasma insulin response to glucose stimulation were the same

for genotypes of each sex, with all mice returning to baseline within two hours of receiving the glucose bolus. (**D**) Insulin curve AUCs were not different. (**E**) C-peptide, a marker of insulin secretion, was the same for genotypes of each sex during the oGTT, with no difference in AUC (**F**). (**G**) The C-peptide/insulin ratio, used as a surrogate for insulin clearance, were not different at 0, 15, and 30 minutes. (H) AUCs for C-peptide/insulin ratio were similar between genotypes of the same sex.
(TIFF)

**S6 Fig. β$_3$-adregeneric receptor agonist stimulation failed to produce a physiologic response in *Abhd2* KO mice.** Plasma glucose concentrations at various time points (**A**) and total AUC for glucose (**B**) during β$_3$-adregeneric receptor agonist stimulation was not different in male or female *Abhd2$^{KO}$* mice. Plasma insulin concentrations (**C**) and total AUC for insulin (**D**) during the B$_3$TT were the same for genotypes of each sex. Non-esterified fatty acid (NEFA) concentration (**E**) and total AUC for NEFA (**F**), and glycerol concentration (**G**), and AUC for glycerol (**H**) during the β$_3$TT did not different for *Abhd2$^{KO}$* female or male mice.
(TIFF)

**S7 Fig. Loss of *Abhd2* does not alter the physiological response to prolonged fasting or refeeding.** Following a 24-hr fast, female mice averaged a 1.7 ± 0.9 gm weight loss and an average 1.1 ± 0.1 gm weight gain following the 6-hour refeed period and were not different for *Abhd2$^{KO}$* versus WT mice (**A**). Plasma NEFAs, measured before and after prolonged fast, were similar between genotypes (**B**). Male mice lost 2.5 ± 0.2 gm with prolonged fasting and regained 0.6 ± 0.1 gm following refeeding and were not different between genotypes (**C**). Plasma NEFAs of male mice during the fast/refeed protocol did not differ by genotype (**D**).
(TIFF)

**S8 Fig. Loss of *Abhd2* did not alter insulin secretion in response to glucose or monoacylglycerol.** Insulin secretion in response to varying glucose concentration, or two different monoacyl-glycerols (2-AG or 1-PG) (**A**) and total islet insulin content (**B**) remained unchanged in cultured islets from female and male *Abhd2$^{KO}$* versus WT mice.
(TIFF)

**S9 Fig. Loss of *Abhd2* exerts a subtle influence on whole-body cholesterol metabolism.** Total plasma cholesterol (**A**) and triglycerides (**B**), biliary cholesterol (**C**), and hepatic cholesterol (**D**) in female and male *Abhd2$^{KO}$* versus WT mice. Male *Abhd2$^{KO}$* mice showed a small increase in hepatic cholesterol (p = 0.06). Plasma cholesterol lipoproteins were separated by FPLC and assayed for cholesterol in female (**E**) and male (**F**) mice. Total AUC for cholesterol in all FPLC fractions (**G**). Liver from female (**H**) and male (**I**) mice were analyzed for LDL-receptor (LDLR) protein content by immunoblot. (**J**) Quantitation of LDLR protein abundance was not different between genotypes of the same sex.
(TIFF)

**S1 Table. 73 protein-coding and non-coding genes are located within a 2Mbp region flanking liver lipid QTL.**
(XLSX)

**S2 Table. 37 lipid classes were detected by untargeted MS-based lipidomics in hepatic tissue of WT and *Abhd2$^{KO}$* mice.**
(XLSX)

**S3 Table. 583 unique lipid species were detected in hepatic tissue of WT and *Abhd2*^KO mice.**
(XLSX)

**S4 Table. QTL of mitochondrial lipids conditioned on sex and wave.**
(XLSX)

**S5 Table. QTL of mitochondrial lipids conditioned on sex, wave, and PC 20:4_22:6.**
(XLSX)

**S6 Table. Internal standard for mass spectrometry-based lipidomics.**
(XLSX)

**S7 Table. Primer sequences for RT-PCR.**
(XLSX)

**S8 Table. Antibodies and source for western blot analyses.**
(XLSX)

**S1 Rawdata. Raw data for *Abhd2* expression in 500 DO and *Abhd2*^KO mouse phenotyping data.**
(ZIP)

## Acknowledgments

We thank Dr. Polina Lishko, currently at Washington University in St. Louis, MO, for providing heterozygous *Abhd2* mice. Lipidomic analysis of mouse liver tissue was performed at the Metabolomics Core Facility at the University of Utah. Mass spectrometry equipment was obtained through NCRR Shared Instrumentation Grant 1S10OD016232-01, 1S10OD018210-01A1 and 1S10OD021505-01. Additional support was provided by the Jackson Laboratory Cube Initiative.

## Author Contributions

**Conceptualization:** Tara R. Price, Gary A. Churchill, Mark P. Keller, Alan D. Attie.

**Data curation:** Tara R. Price, Donnie S. Stapleton, Gary A. Churchill, Mark P. Keller.

**Formal analysis:** Tara R. Price, Brian S. Yandell, Mark P. Keller, Alan D. Attie.

**Funding acquisition:** Gary A. Churchill, Mark P. Keller, Alan D. Attie.

**Investigation:** Tara R. Price, Donnie S. Stapleton, Marie K. Norris, Brian W. Parks, William L. Holland, Mark P. Keller, Alan D. Attie.

**Methodology:** Tara R. Price, Kathryn L. Schueler, Marie K. Norris, Brian W. Parks, Brian S. Yandell, William L. Holland, Mark P. Keller, Alan D. Attie.

**Project administration:** Mark P. Keller, Alan D. Attie.

**Resources:** Kathryn L. Schueler, Marie K. Norris, Brian W. Parks, Gary A. Churchill, William L. Holland, Mark P. Keller, Alan D. Attie.

**Software:** Gary A. Churchill, Mark P. Keller.

**Supervision:** Mark P. Keller, Alan D. Attie.

**Validation:** Tara R. Price, Mark P. Keller, Alan D. Attie.

**Visualization:** Tara R. Price, Brian S. Yandell, Mark P. Keller, Alan D. Attie.

**Writing – original draft:** Tara R. Price, Mark P. Keller, Alan D. Attie.

**Writing – review & editing:** Tara R. Price, Brian S. Yandell, Gary A. Churchill, William L. Holland, Mark P. Keller, Alan D. Attie.

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
