## [Decision Letter · Decision Letter 0]

30 Apr 2023

Dear Dr Attie,

Thank you very much for submitting your Research Article entitled 'Lipidomic QTL in Diversity Outbred mice identifies a novel function for α/β hydrolase domain 2 (*Abhd2*) as an enzyme that metabolizes phosphatidylcholine and cardiolipin' to PLOS Genetics.

The manuscript was fully evaluated at the editorial level and by independent peer reviewers. The reviewers appreciated the attention to an important topic but identified some concerns that we ask you address in a revised manuscript.

We therefore ask you to modify the manuscript according to the review recommendations. Your revisions should address the specific points made by each reviewer.

Yours sincerely,

David A. Buchner, PhD

Guest Editor

PLOS Genetics

Scott Williams

Section Editor

PLOS Genetics

Reviewer's Responses to Questions

**Comments to the Authors:**

Reviewer #1: This study by Price et al identifies Abhd2 as a susceptibility gene for phospholipid metabolism in the liver of mice via QTL analysis, which is subsequently validated by findings in a Abhd2 mouse model. This is a relatively straightforward paper with logical progression of hypothesis and analysis. The mass spectrometry based lipidomics for a large cohort is striking and the role of Abhd2 in modulating phospholipid is well demonstrated. However, a few concerns or suggestions are noted to improve the rigor of the results and conclusions.

1) LOD threshold: Permutation tests are generally performed to determine the threshold for significant or suggestive linkage, especially when multiple covariates (sex, wave, and PC-20:4/22:6) are included. In the current study, the LOD score threshold for QTL identification was arbitrarily set at 6. Is this threshold for significant or suggestive linkage?

2) Mediation analysis to infer causality: It is stated “Mediation analysis to establish causality was performed using conditional regression of the target phenotype on gene expression of candidate gene and the locus genotype”, but results after adjustment with “the locus genotype” have not shown.

3) The genetic background of Abhd2 knockout mice was not provided.

4) Plasma cholesterol was measured, but why not phospholipids?

5) The method for measuring mitochondrial lipids should be provided.

Reviewer #2: This study demonstrates a thorough investigation of the role of Abhd2 in hepatic phospholipid regulation using a combination of genetic association, gene expression, lipidomics, and validation experiments.

I have minor comments:

1. The study could benefit from a more detailed discussion of the limitations, such as the lack of mechanistic insight into how Abhd2 regulates phospholipid levels and the need for further investigation into tissue-specific roles of Abhd2 in lipid metabolism.

2. Given that Abhd2 has been identified as a candidate gene in human GWAS studies (https://www.ebi.ac.uk/gwas/genes/ABHD2) for lipid-related diseases, such as age-related macular degeneration (as reported in the paper by Persad et al., 2017), it would be valuable for the authors to discuss the potential relevance of their findings to these diseases.

Persad PJ, Heid IM, Weeks DE, Baird PN, de Jong EK, Haines JL, Pericak-Vance MA, Scott WK; International Age-Related Macular Degeneration Genomics Consortium (IAMDGC). Joint Analysis of Nuclear and Mitochondrial Variants in Age-Related Macular Degeneration Identifies Novel Loci TRPM1 and ABHD2/RLBP1. Invest Ophthalmol Vis Sci. 2017 Aug 1;58(10):4027-4038. doi: 10.1167/iovs.17-21734. PMID: 28813576; PMCID: PMC5559178.

3. Consider mentioning the mouse reference genome build used in the study to ensure reproducibility and clarity for future research.

4. I suggest including more details about how the threshold for significance was determined for R/QTL2 analysis. This could include information about the rationale behind the choice of a LOD score greater than 6.0 as the threshold for identifying a QTL

Reviewer #3: The manuscript by Price and colleagues integrates genomic association with expression data leading to the identification of Abhd2 as a driver of hepatic phospholipid phenotypes in mice.

Figure 1B: It will facilitate the understanding of the readers if the authors write the genotypes of the alleles versus Abhd2 mRNA and phospholipid levels.

Line 124, can the authors include a sentence about the significance of the observation. Something like “genetic architecture, suggesting that Abhd2 could participate in PCs degradation”.

Line 132: Please explain what mediation analysis is. This will facilitate the understanding of the readers.

Figure 2 could be supplementary. It supports the findings observed in Fig 1.

Figure 3: Lipidomics in Abhd2 ko

In what genetic background is the Abdh2 ko, and how is Abhd2 expressed in this strain? This aspect is relevant because the consequences of its deletion could be different if it is in a highly expressed tissue or a low expression tissue.

In what subcellular compartments are these lipids stored? Do the Abhd2 ko show features of a Lysosomal Storage Disorder (LSD)? Could Abhd2 be a modifier gene of phospholipidosis? Can the authors show electron microscopy images of the tissues or discuss this point?

Gene expression of mitochondrial genes does not say anything about its functionality. Can the authors show any functional data of the mitochondria?

Can the authors expand the discussion about the sex-specific lipid changes in the Abdh2 ko? Does the Abdh2 promotor has an estrogen-responding element? What is the sex of the animals where the initial screening was performed?

Figure 4:

Figure 1B: it is difficult to interpret the results how they are currently plotted. Can the authors show the correlations between the gene expression of Abh2 and Abh18 genes across the different DO parental strains? Do Abh2 and Abh18 co-localized? Please show a confocal image to support that they could be working together. Is any data predicting their physical interactions?

In vivo physiological measurements:

Did the authors measure liver function tests, such as alanine transaminase (ALT) and aspartate transaminase (AST), alkaline phosphatase (ALP), gamma-glutamyl transferase (GGT), among others?

**Have all data underlying the figures and results presented in the manuscript been provided?**

Reviewer #1: Yes

Reviewer #2: Yes

Reviewer #3: Yes

PLOS authors have the option to publish the peer review history of their article (what does this mean?). If published, this will include your full peer review and any attached files.

Reviewer #1: No

Reviewer #2: No

Reviewer #3: **Yes: **Andres D. Klein

---

## [Decision Letter · Decision Letter 1]

3 Jul 2023

Dear Dr Attie,

We are pleased to inform you that your manuscript entitled "Lipidomic QTL in Diversity Outbred mice identifies a novel function for α/β hydrolase domain 2 (*Abhd2*) as an enzyme that metabolizes phosphatidylcholine and cardiolipin" has been editorially accepted for publication in PLOS Genetics. Congratulations!

Yours sincerely,

David A. Buchner, PhD

Guest Editor

PLOS Genetics

Scott Williams

Section Editor

PLOS Genetics

Comments from the reviewers (if applicable):

Reviewer's Responses to Questions

**Comments to the Authors:**

Reviewer #1: The authors have adequately addressed my concerns.

Reviewer #2: The authors have adequately addressed all my previous comments in the revised manuscript. I have no further comments or suggestions.

Reviewer #3: well done

**Have all data underlying the figures and results presented in the manuscript been provided?**

Reviewer #1: Yes

Reviewer #2: Yes

Reviewer #3: Yes

PLOS authors have the option to publish the peer review history of their article (what does this mean?). If published, this will include your full peer review and any attached files.

Reviewer #1: No

Reviewer #2: No

Reviewer #3: **Yes: **Andrés D Klein

**Data Deposition**

http://datadryad.org/submit?journalID=pgenetics&manu=PGENETICS-D-23-00321R1

**Press Queries**

---

## [Editor Report · Acceptance letter]

26 Jul 2023

PGENETICS-D-23-00321R1 

Lipidomic QTL in Diversity Outbred mice identifies a novel function for α/β hydrolase domain 2 (*Abhd2*) as an enzyme that metabolizes phosphatidylcholine and cardiolipin 

Dear Dr Attie, 

We are pleased to inform you that your manuscript entitled "Lipidomic QTL in Diversity Outbred mice identifies a novel function for α/β hydrolase domain 2 (*Abhd2*) as an enzyme that metabolizes phosphatidylcholine and cardiolipin" has been formally accepted for publication in PLOS Genetics! Your manuscript is now with our production department and you will be notified of the publication date in due course.

With kind regards,

Zsofia Freund

PLOS Genetics

On behalf of:
